# Optimizing the deployment of ultra-low volume and targeted indoor residual spraying for dengue outbreak response

**Sean M. Cavany**[1]*, **Guido España**[1], **Alun L. Lloyd**[2], **Lance A. Waller**[3], **Uriel Kitron**[4], **Helvio Astete**[5], **William H. Elson**[6], **Gonzalo M. Vazquez-Prokopec**[4], **Thomas W. Scott**[6], **Amy C. Morrison**[7], **Robert C. Reiner Jr.**[8], **T. Alex Perkins**[1]*

1 Department of Biological Sciences & Eck Institute of Global Health, University of Notre Dame, Notre Dame, Indiana, United States of America, 2 Department of Mathematics & Biomathematics Graduate Program, North Carolina State University, Raleigh, North Carolina, United States of America, 3 Department of Biostatistics and Bioinformatics, Rollins School of Public Health, Emory University, Atlanta, Georgia, United States of America, 4 Department of Environmental Sciences, Emory University, Atlanta, Georgia, United States of America, 5 U.S. Naval Medical Research Unit Six, Lima, Peru, 6 Department of Entomology and Nematology, University of California, Davis, Davis, California, United States of America, 7 Department of Pathology, Microbiology, and Immunology, School of Veterinary Medicine, University of California, Davis, Davis, California, United States of America, 8 Institute of Health Metrics and Evaluation, University of Washington, Seattle, Washington, United States of America

* scavany@nd.edu (SMC); taperkins@nd.edu (TAP)

**Data Availability Statement:** Data and code are available at https://github.com/scavany/dengue_outbreak_response.

## Abstract

Recent years have seen rising incidence of dengue and large outbreaks of Zika and chikungunya, which are all caused by viruses transmitted by *Aedes aegypti* mosquitoes. In most settings, the primary intervention against *Aedes*-transmitted viruses is vector control, such as indoor, ultra-low volume (ULV) spraying. Targeted indoor residual spraying (TIRS) has the potential to more effectively impact *Aedes*-borne diseases, but its implementation requires careful planning and evaluation. The optimal time to deploy these interventions and their relative epidemiological effects are, however, not well understood. We used an agent-based model of dengue virus transmission calibrated to data from Iquitos, Peru to assess the epidemiological effects of these interventions under differing strategies for deploying them. Specifically, we compared strategies where spray application was initiated when incidence rose above a threshold based on incidence in recent years to strategies where spraying occurred at the same time(s) each year. In the absence of spraying, the model predicted 361,000 infections [inter-quartile range (IQR): 347,000–383,000] in the period 2000–2010. The ULV strategy with the fewest median infections was spraying twice yearly, in March and October, which led to a median of 172,000 infections [IQR: 158,000–183,000], a 52% reduction from baseline. Compared to spraying once yearly in September, the best threshold-based strategy utilizing ULV had fewer median infections (254,000 vs. 261,000), but required more spraying (351 vs. 274 days). For TIRS, the best strategy was threshold-based, which led to the fewest infections of all strategies tested (9,900; [IQR: 8,720–11,400], a 94% reduction), and required fewer days spraying than the equivalent ULV strategy (280). Although spraying twice each year is likely to avert the most infections, our results

**Funding:** SMC, GE, GMV-P, ACM, TWS, RCR, and TAP were supported by grant P01AI098670 (TWS, PI) from the National Institutes of Health, National Institute for Allergy and Infectious Disease (https://www.niaid.nih.gov). The funders had no role in the study design, data collection and analysis, decision to publish, or preparation of the manuscript.

**Competing interests:** The authors have declared that no competing interests exist.

indicate that a threshold-based strategy can become an alternative to better balance the translation of spraying effort into impact, particularly if used with a residual insecticide.

## Author summary

Over half of the world's population is at risk of infection by dengue virus (DENV) from *Aedes aegypti* mosquitoes. While most infected people experience mild or asymptomatic infections, dengue can cause severe symptoms, such as hemorrhage, shock, and death. A vaccine against dengue exists, but it can increase the risk of severe disease in people who have not been previously infected by one of the four DENV serotypes. In many places, therefore, the best currently available way to prevent outbreaks is by controlling the mosquito population. Our study used a simulation model to explore alternative strategies for deploying insecticide in the city of Iquitos in the Peruvian Amazon. Our simulations closely matched empirical patterns from studies of dengue's ecology and epidemiology in Iquitos, such as mosquito population dynamics, human household structure, demography, human and mosquito movement, and virus transmission. Our results indicate that an insecticide that has a long-lasting, residual effect will have the biggest impact on reducing DENV transmission. For non-residual insecticides, we find that it is best to begin spraying close to the start of the dengue transmission season, as mosquito populations can rebound quickly and resume previous levels of transmission.

## Introduction

Dengue incidence is rising [1–3]. Current estimates indicate that over half of the world's population is at risk of dengue virus infection (DENV) [4]. The last decade has also seen explosive outbreaks of Zika and chikungunya viruses, which are transmitted by *Aedes aegypti*, too. Because the only licensed dengue vaccine is contraindicated in individuals without prior DENV exposure [5], and there are no therapeutic options for Zika and chikungunya, the only intervention available to address these diseases in most settings is vector control. The most common method for controlling adult *Ae. aegypti* is ultra-low volume (ULV) spraying, defined as a treatment with minimum effective volume of the active ingredient [6–8]. It can be implemented outdoors by plane [9] or trucks [10], or indoors by handheld devices. Indoor application is considered most effective, because *Ae. aegypti* lives primarily inside human habitations [6,11]. In Iquitos, Peru, where our study was focused, ULV is the most commonly used method and has been repeatedly applied city-wide in response to past *Aedes*-transmitted virus outbreaks [12].

Apart from the Western hemisphere-wide *Ae. aegypti* control program, which focused on yellow fever prevention during the 1950s and 1960s [13], there have been two vector control programs that have successfully controlled dengue: Cuba, which used ULV spraying complemented by larval source reduction [14], and Singapore, which utilized larval source reduction and community engagement [15]. A 2010 systematic review found five studies on indoor ULV [6], with generally high (up to 100%) mosquito mortality effects that were sustained for only about one month [16–19]. An exception was a study in Thailand, which found a sustained drop in *Ae. aegypti* landing rates out to six months [20]. A more recent 2016 systematic review found no randomized controlled trials assessing the impact of ULV spraying. A recent study in Iquitos reported that city-wide indoor ULV spraying reduced the *Ae. aegypti* population by

60%, but effects were only sustained for a short period, also about one month [11]. The sum of all this evidence indicates that indoor ULV spraying is effective at reducing adult mosquito numbers in the short term [16–18], but with mixed evidence on its impact on virus transmission and disease [6,21,22] and a lack of information on best practices for how to deploy ULV at a city level [21,23].

Traditionally, indoor residual spraying (IRS) has been widely used against malaria, but has not been recommended for control of *Aedes*-borne diseases [6]. In recent years, however, there has been increased interest in utilizing IRS to combat *Aedes*-borne diseases. A 2016 systematic review found no evidence of an effect of IRS on DENV infection risk [21], though only two studies were included in that analysis. A more recent study using contact tracing reported a large epidemiological effect, reducing the probability of future transmission by 86–96% in Cairns, Australia [24]. A study of IRS in Iquitos found more than 80% mosquito mortality in 24 hours for eight weeks after spraying [12]. Recent work to develop targeted IRS (TIRS), where insecticide is sprayed only where *Ae*. a*egypti* are likely to rest, led to gains in speed of application, without significant declines in effectiveness [25]. Increased speed coupled with the relatively small size of Iquitos makes it a feasible location to undertake city-wide TIRS spraying. The primary drawback is that, there are no published details on best practices for undertaking this approach.

Field trials to measure effectiveness and compare different strategies are logistically challenging and in some cases prohibitively expensive due to the complex interplay of mosquito population dynamics, seasonal dynamics, human movement, and fine-scale heterogeneities [21,26–34]. Mathematical modeling can be helpful in multifaceted cases like this for predicting the best intervention strategies. Additionally, the rebound in adult mosquito abundance following spraying [11,18,21], due to immature emergence and movement, and feedbacks caused by reduced egg-laying due to increased adult mortality, mean that it is important to capture mosquito population dynamics when modeling vector control strategies. For example, models have been used for many diseases to analyze causes of outbreaks and to help optimize response strategies; increasingly, this is happening in real-time during outbreaks [35]. Examples include diphtheria among Rohingya refugees [36], where real-time modeling informed resource allocation and transmission mechanisms; the 2013–16 west African Ebola outbreak [37,38]; optimum vaccination strategies in response to measles outbreaks [39–42]; and seasonal malaria prophylaxis [43,44]. Recent modeling studies evaluated the impact on dengue of outdoor, truck-mounted ULV spraying in Porto Alegre, Brazil and IRS in Merida, Mexico [10,45]. The former reported that 24% of cases were averted and the latter found that IRS strategies initiated early in the transmission season were generally superior to those initiated late in the season.

One challenge associated with ULV and IRS campaigns is determining the criteria for initiating such a response. Brady et al. [46] discussed a variety of ways to determine when a dengue outbreak is occurring, predominantly based on comparison of current incidence from patterns in recent years. An alternative to initiating an intervention when a threshold has been exceeded would be to start the intervention at the same time each year in an effort to prevent transmission from reaching outbreak levels. Hladish et al., considered campaign start date for IRS, finding that deploying IRS four months before the seasonal peak produced the greatest impact on infections [45]. Several studies of malaria also found that IRS timing was important [44,47], and one study assessed ULV timing in relation to *Triatoma dimidiata*, the vector for Chagas disease [48]. Few studies have compared alternative methods for initiating outbreak response though, and none, to our knowledge, did so for the impact of indoor ULV on dengue. An added complication is the characteristic variation in seasonal patterns of DENV transmission [22], which along with the aforementioned complex interplay of heterogeneities can result

in vector control strategies with the biggest impact on mosquitoes not necessarily corresponding to the biggest reduction in human infections.

To address these challenges, in this study we used a transmission model to investigate the optimal application of indoor ULV or TIRS for dengue control in Iquitos, Peru. Because the timing of DENV transmission seasons can vary considerably across years, we sought criteria that were optimal in the sense of being robust across multiple years, rather than optimizing outbreak response for a single outbreak year. We compared several possible threshold-based strategies, based on a variety of outbreak definitions, to strategies in which insecticide was sprayed regularly on the same date, either once or twice a year, starting at different times each year.

## Methods

### Ethics statement

The study protocol was approved by the Naval Medical Research Unit No. 6 (NAMRU-6) Institutional Review Board (IRB) (protocol #NAMRU6.2014.0028), in compliance with all applicable Federal regulations governing the protection of human subjects. IRB relying agreements were established between NAMRU-6, the University of California, Davis, Tulane University, Emory University and Notre Dame University. The protocol was reviewed and approved by the Loreto Regional Health Department, which oversees health research in Iquitos. This study represents historical data analysis using data without personal identifiers.

### Study area and synthetic location generation

Our model was calibrated to data from Iquitos, which has a population of about 450,000 people in the Peruvian Amazon [49,50] and where all four DENV serotypes are endemic. We calibrated the initial level of population immunity, number of imported infections, and the absolute mosquito abundance using a particle filtering approach to an estimate of the serotype-specific number of infections based on a longitudinal cohort study in Iquitos [51]. Results of this calibration procedure are shown in S1 Fig, S2 Fig and S3 Fig, including a comparison of model predicted numbers of infections and the estimated number to which we calibrated in S1 Fig. Full details of this calibration procedure are given in Perkins et al. [49]. Several other components of the model were independently calibrated to data from Iquitos, including the spatio-temporal pattern in mosquito abundance [52] and human movement [53]. Our analysis focuses on the period 2000–2010, in which DENV-3 and DENV-4 were introduced (in 2001 and 2008, respectively). During this period, household spraying efforts were ongoing in Iquitos [52]. The spatio-temporal pattern in mosquito abundance was calibrated to a counterfactual surface, which removed the effect of this spraying, though it was present in the particle filter calibration of the number of imported cases. Locations and coordinates of almost half (40,839/ 92,891) of the locations in the city were collected during surveys conducted as part of prospective cohort studies [52]. For the remaining locations, we randomly assigned them to ministry of health zones, so that the total number within each zone matched that recorded in past city-wide spraying campaigns [11]. The location type (e.g., home, shop, etc.) of each of these new locations was randomly assigned so that the final distribution of location types matched that from the aforementioned surveys. Their positions were distributed using the rSSI algorithm in the spatstat package in R [54,55], so that they were evenly distributed, and at least 5m separated each location.

## Model overview

We simulated outbreak response strategies using an established agent-based model of dengue virus dynamics in Iquitos (Fig 1B). This model has been shown to accurately recreate the dynamics of all four DENV serotypes in Iquitos, and has previously been used to answer questions relating to DENV vaccination [49,50,56]. Human agents in the model move according to realistic movement patterns in Iquitos [53]. Household composition and demographic patterns match those seen in Iquitos and Peru as a whole, respectively. Mosquito agents move with fixed probability of 0.3 to a nearby location [57] and have a propensity to bite that depends on temperature, the host's body size, and whether it is the mosquito's first bite. Four stages of mosquito development are explicitly modeled (eggs, larvae, pupae, and female adults), with density-dependent mortality occurring in the larval stage. Mosquito population dynamics were calibrated, via an additional mortality rate acting on pupae and larvae, so that adult female abundance matched a spatiotemporal estimate of abundance in Iquitos [52]. The model assumed that all four DENV serotypes can be transmitted when either a mosquito or human is infectious, the other susceptible, and the mosquito takes a bloodmeal. Transmission occurs with probability 0.9 from mosquitoes to humans and a time-varying probability from humans to mosquitoes [58,59]. Following infection with one DENV serotype, human agents exhibit permanent immunity to that serotype and temporary immunity to the other serotypes for a period of 686 days on average [60]. The rate of introduction of each DENV serotype into the population was calibrated so that serotype-specific incidence of infection matched that predicted for Iquitos in a previous study [51]. All features of the model have been thoroughly described in a prior publication [49], and further details are described in the S1 Text.

## Hypothetical spraying protocol

We set in place an outbreak response intervention based on a zonal spraying strategy. Spraying takes place on Monday through Saturday. There are 35 Ministry of Health zones in Iquitos, and the outbreak response sprays these 35 zones in batches of a fixed number, until all zones are sprayed (Figs 1A and 2). In the baseline scenario, we start in zone 1 and spray zones sequentially, though we explore other orderings in a sensitivity analysis. Immediately after all zones are sprayed, another cycle of spraying the 35 zones in batches is initiated, and this process is repeated until a fixed number of cycles have been completed (three cycles for ULV, or one cycle for TIRS). We explore the effect of leaving a gap between spray cycles in the sensitivity analysis. The number of houses to spray per day is limited by a maximum number of houses that can be sprayed each day. The probability that occupants will be at home and allow the outbreak response team to spray is represented by a compliance probability. The form of vector control is assumed to be an adult insecticide that increases the baseline mortality of mosquitoes by a fixed hazard, called thoroughness. For ultra-low volume (ULV) spraying, the increase in mortality decays exponentially following spraying with a half-life of one day. The most realistic parameterization, based on internal US Naval records of past city-wide spraying campaigns that took place in 2014, involved attempting to spray 11,000 houses per day on average, of which ~6,800 (62%) were compliant. Six or seven zones were sprayed simultaneously so that each zone was sprayed three times over a 3–4 week period. There was no training period, no waiting periods between spray cycles, and no repeat visits to houses. We calibrated the intervention thoroughness such that the intervention campaign would generate an approximately 60% drop in total city-wide mosquito abundance following spraying, consistent with empirical estimates for Iquitos by Gunning et al. [11]. Spraying parameters are given in Table 1).

We also simulated city-wide TIRS spraying. In this case, we changed the number of houses sprayed per day, the thoroughness (i.e., the increase in mosquito mortality), and the residuality

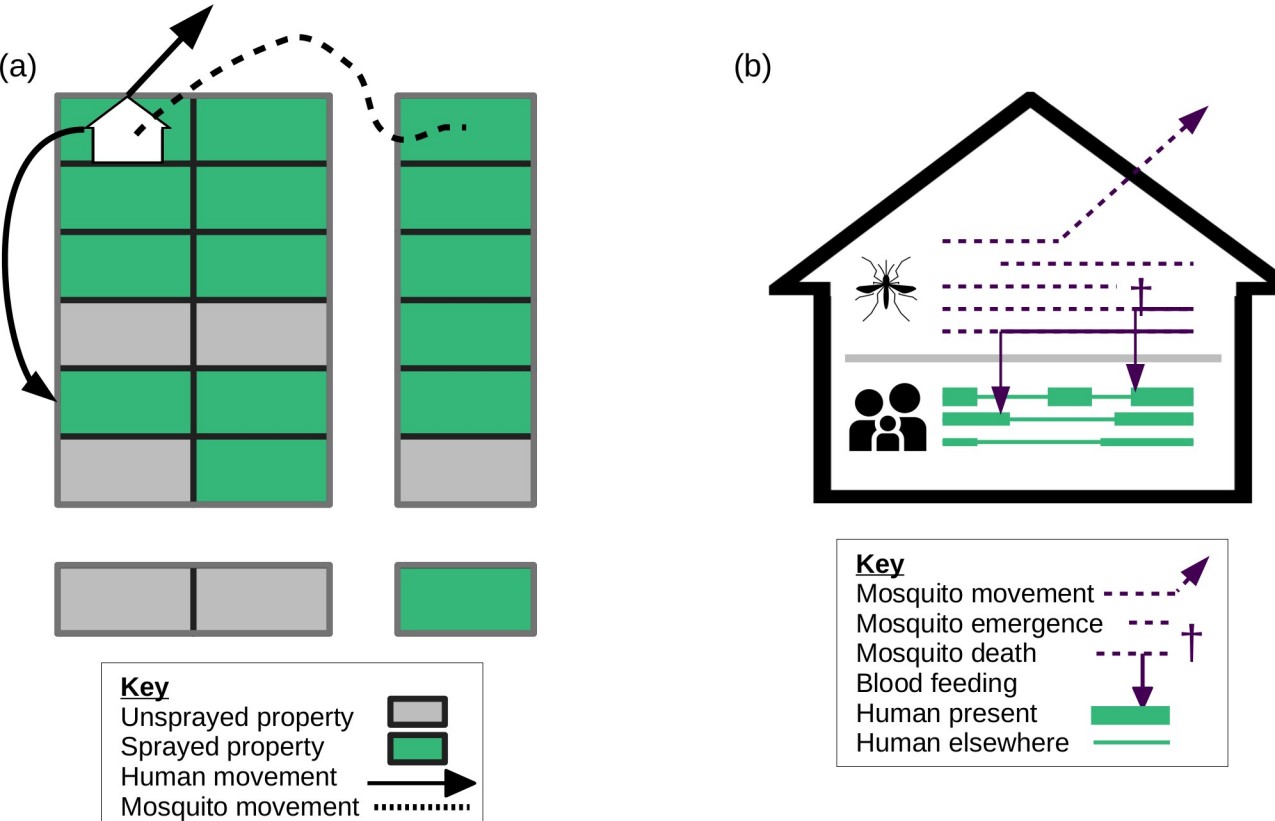

**Fig 1.** (a) Example of a block of houses that were sprayed. 70% of houses are compliant and were sprayed in the baseline. Both humans and mosquitoes moved between locations. Mosquitoes moved to nearby locations, whereas humans moved around the city according to a movement model calibrated to a location survey in Iquitos. (b) On any given day, mosquitoes could move, emerge, die, or take a bloodmeal. Humans had different levels of attractiveness to mosquitoes based on their body size (represented here by line-width). Icons in this figure are taken from thenounproject.com.

of the insecticide. Based on the estimate that it takes 5–6 times longer to spray a house using TIRS compared to ULV (~3 minutes vs. ~15 minutes) [25], we used 2,000 as an upper limit on the number of houses sprayed daily for TIRS. We calibrated the thoroughness and the residuality so that the 24-hour mortality matched that observed for TIRS in Dunbar et al. [25]. This led to a function which had an increase in mosquito mortality of nine deaths/mosquito-day (i.e., increased the daily risk to close to 1), which decayed exponentially after 90 days following treatment, with a half-life of 11 days (Fig 3). In the TIRS scenario, each campaign consisted of just one city-wide cycle, compared to three spray cycles for ULV campaigns.

## Experiments

We considered three ways in which spraying could be initiated: when incidence exceeds a threshold, once yearly, or twice yearly. In the threshold-based strategies, spraying is initiated when the weekly or monthly incidence rises one or two standard deviations above the mean incidence from the corresponding week or month from the previous five years (henceforth, adaptive threshold strategies), or when weekly or monthly incidence rises above a fixed threshold (henceforth, fixed threshold strategies) [46]. This leads to a total of four possible adaptive threshold strategies. Note that for the purposes of initiating threshold strategies, incidence represents cases that are symptomatic, whereas in the results, we generally report the number of infections a particular strategy leads to, irrespective of symptoms. The yearly and twice yearly

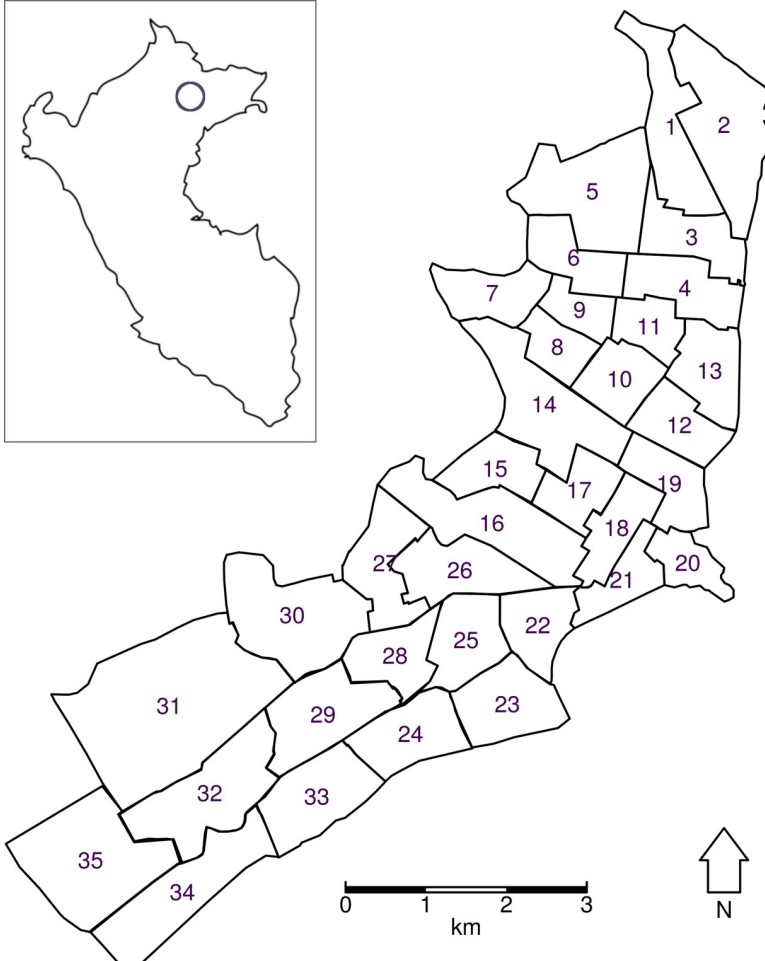

**Fig 2. Map of Iquitos showing the boundaries of the Ministry of Health zones, which are numbered 1–35.** Inset shows location of Iquitos in Peru. In the baseline scenario spraying begins in zone 1 and continues in ascending order through the zones. The inset map in this figure is taken from thenounproject.com.

strategies begin at the same time(s) each year. We tested yearly spraying starting in each month (12 strategies) and twice-yearly spraying in each pair of months (66 strategies) (Table 1). Due to the residual effect of TIRS and the longer roll-out of the campaign we did not explore twice-yearly strategies for this intervention. We compared the number of infections predicted under each of these initiation strategies to the number predicted had there been no spraying over the years 2000–2010. To focus on the effect of the strategy for initiating spraying, we sprayed the Ministry of Health zones in the same order in each simulation. For the same reason, we used the same time series of DENV introduction in each simulation; namely, the trajectory associated with the highest likelihood following the calibration procedure. We chose the number of simulations so that in the absence of spraying, the change in the coefficient of variation of the number of human infections as new simulations were added was less than 0.1% (about 400 simulations) [61]. For model outputs we present the median and the inter-quartile range (IQR). We use the IQR as the model is highly stochastic and this measure of dispersion is robust to the presence of outliers.

**Table 1. Summary of simulation experiments.** Thoroughness refers to the initial increase in mosquito mortality in sprayed houses. In the case of TIRS this begins to decay with the given half-life after the length of full effect.

| Strategy | Ultra-low volume | Targeted indoor residual spraying |
|---|---|---|
| Adaptive threshold | Based on mean and standard deviation from recent years. **4 strategies, 400 simulations each.** | |
| Fixed threshold | Vary threshold between 1 and 1,000 per month, and 1 and 230 per week. **2,000 simulations.** | |
| Yearly | Start at the beginning of each month. **12 strategies, 400 simulations each.** | |
| Twice yearly | All pairs of months. **66 strategies, 400 simulations each.** | N/A |
| Baseline spraying parameter values (explored ranges) | | |
| Thoroughness | 1.5 /day (0.5/day, 10/day) | 9 /day (0.5/day, 10/day) |
| Length of full effect | 1 day | 90 days |
| Half-life | 0 | 11 days |
| Delay between cycles | 0 (0 days, 31 days) | 0 |
| Compliance of households | 70% (30%, 100%) | 70% (30%, 100%) |

## Sensitivity analysis

For each of the optimum adaptive threshold, yearly, and, in the case of ULV, twice yearly strategies, we undertook a global sensitivity analysis of the total number of human infections and mosquito abundance. For each of the parameters governing spraying (thoroughness, delay between cycles, for ULV only, and compliance), we selected a range of plausible values using the sampling approach of Saltelli et al. [62] (Table 1), and simulated the best outbreak response strategies for each of these [63]. We then decomposed the variance in the output into first order effects of the sampled parameters using the SALib package in Python [64]. Because the data used to parameterize the TIRS strategy were from a controlled experiment, we also reduced the thoroughness in the TIRS adaptive threshold strategy to the value used for ULV spraying (1.5) and to half this value (0.75), while keeping all other parameters the same. Finally, we simulated the adaptive threshold strategies for both IRS and ULV in scenarios where (a) only 10% of cases were reported, and (b) there was a lag of 2 weeks between infection and notification. These scenarios were chosen to explore how under-reporting or reporting delays would affect the adaptive threshold strategies, and only relate to when and whether infections are counted towards the threshold for spraying. They do not affect the model calibration. Moreover, as the model was calibrated to serotype-specific estimates of infections based on serology from longitudinal cohort studies, the calibration was unaffected by surveillance underreporting and delays [49,51].

## Results

Unsurprisingly, TIRS was able to prevent more infections overall than ULV (Table 2). The best adaptive threshold strategies for TIRS started more quickly following a rise in incidence than the best ULV strategies and earlier in the year for the best yearly strategy. Due to its higher efficacy and long-lasting effect, TIRS had an order of magnitude greater impact than ULV on the number of infections predicted.

## Ultra-low volume spraying

In the absence of spraying, the model predicted a median of 361,000 infections [IQR: 347,000–383,000] across the four serotypes in Iquitos in the period 2000–2010. The adaptive threshold

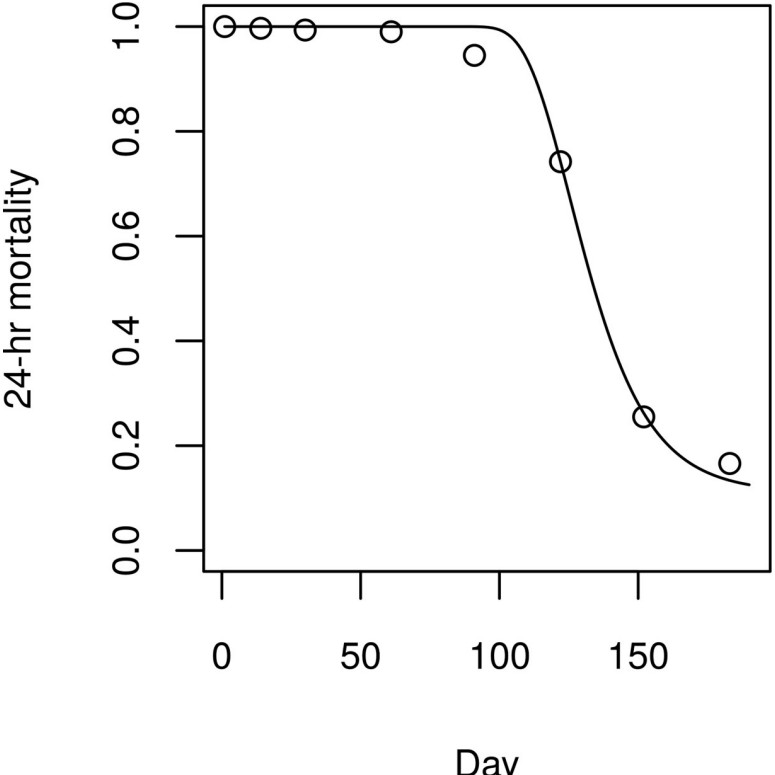

**Fig 3. Mortality over time following TIRS spraying.** Circles represent data from Dunbar et al. [25], line represents mortality function fitted by least squares ($R^2 = 0.995$).

strategy performed best when the incidence was monitored on a monthly basis and when spraying was initiated when incidence exceeded the mean plus one standard deviation from the last five years (254,000 infections; IQR: 210,000–277,000). The difference in the impact on incidence between the adaptive threshold strategies was small (Fig 4A). When spraying was initiated yearly, our model predicted that starting in September would lead to the fewest infections (261,000; IQR: 250,000–277,000), although spraying in October produced slightly higher, but similar results (262,000; IQR: 253,000–275,000) (Fig 4B). In the case of yearly spraying, timing was important; we saw large differences between the best and worst strategies. It is also worth noting that the yearly strategy which led to the lowest average mosquito abundance was spraying in November, not September (S4 Fig). The best strategy for spraying twice yearly was spraying in September and November (172,000; IQR: 158,000–183,000) (Fig 4C). Generally,

**Table 2. Summary of results and predicted number of infections.** N/A denotes not applicable, numbers in cells represent the number of infections over the 11 year period in 1,000s, and the numbers in brackets represent the inter-quartile range (IQR).

| | | None | Adaptive threshold | Yearly | Twice yearly |
|---|---|---|---|---|---|
| ULV | Best strategy | N/A | When monthly incidence is 1σ above mean | September | September & November |
| | Number of infections, 2000–2010 (1,000s) [IQR] | 361 [347, 383] | 254 [210, 277] | 261 [250, 277] | 172 [158, 183] |
| | Days spent spraying [IQR] | 0 | 351 [341, 371] | 274 [272, 277] | 549 [545, 552] |
| TIRS | Best strategy | N/A | When weekly incidence is 1σ above mean | September | N/A |
| | Number of infections, 2000–2010 (1,000s) [IQR] | 361 [347, 383] | 9.90 [8.72, 11.3] | 11.8 [9.75, 14.1] | N/A |
| | Days spent spraying [IQR] | 0 | 280 [175, 315] | 385 [385, 385] | N/A |

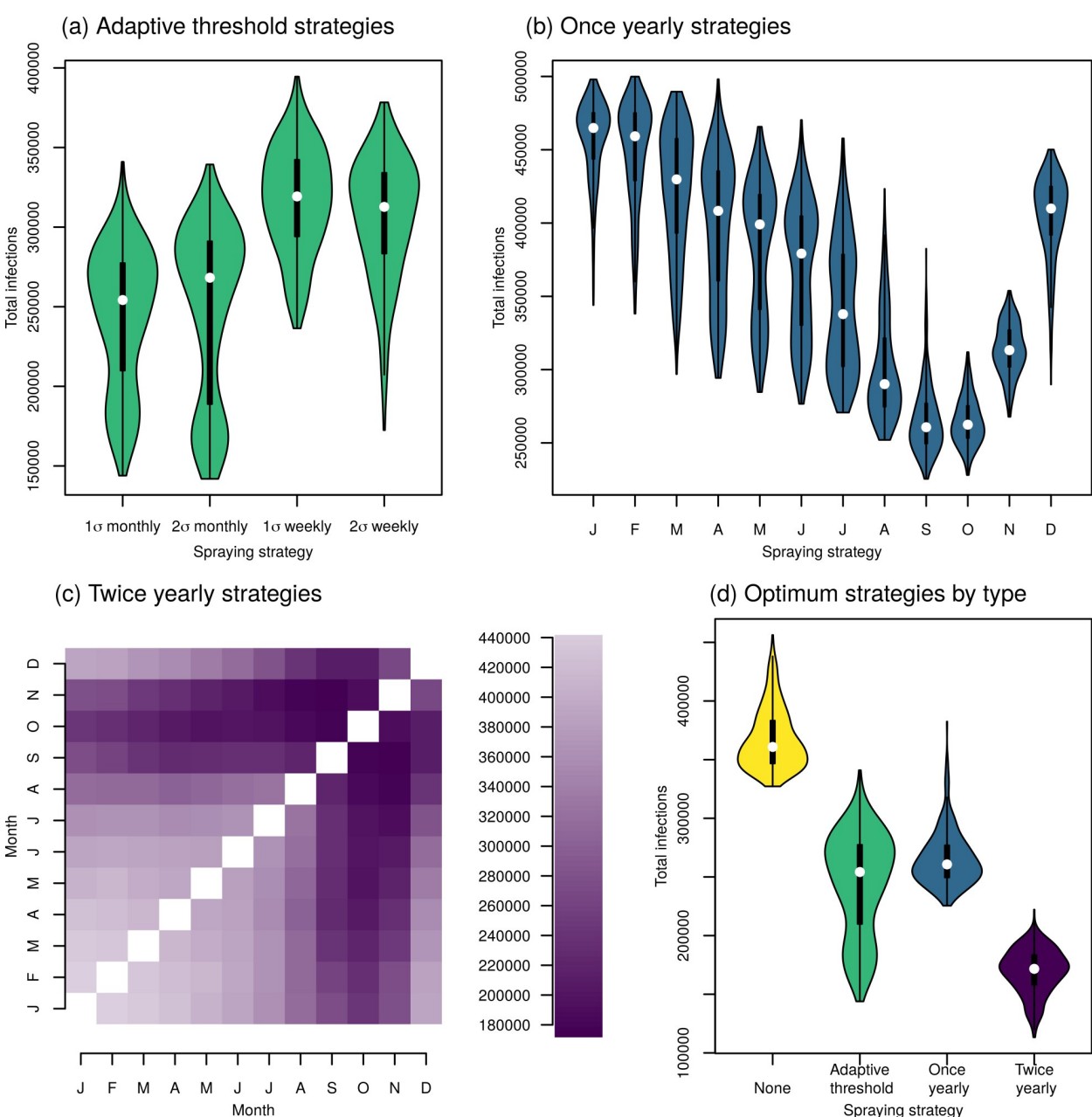

**Fig 4. Predicted human infections following city-wide ULV spraying.** (a) Comparison of adaptive threshold strategies for initiating spraying; spraying began when the monthly or weekly incidence was one or two standard deviations above the mean for that period from the last five years, as shown on the x-axis. (b) Comparison of yearly city-wide spraying strategies, beginning on the first day of the shown month. (c) Comparison of the median predicted infections for twice yearly spraying strategies, beginning on the first days of the shown month. Darker colors correspond to fewer infections, and the diagonal shows yearly spraying strategies. (d) Comparison of the best strategies in each category: adaptive threshold corresponds to starting when monthly incidence was more than one standard deviation above the mean, once yearly corresponds to spraying in September, twice yearly corresponds to spraying in September and November.

undertaking the first spray in August or September (typically, just before the dengue season) and the second in October or November (typically, near the start of the dengue season) led to the fewest infections (Fig 4C). In this case, the strategy that led to the fewest infections also led to the lowest average mosquito abundance: spraying in September and November (S4 Fig).

Comparing the best strategies, spraying twice yearly (in October and March) averted the most infections, but required the most spraying campaigns: 22 campaigns of spraying, spending a median of 549 days spraying in total over 11 years. The best adaptive threshold strategy typically led to fewer infections than the best yearly strategy, but required more spray campaigns (a median of 14 [IQR: 14–15] compared to 11 for a yearly strategy). In seasons with a large outbreak (2000–01, 2001–02, 2002–03, 2008–09, and 2009–10), the adaptive threshold strategy typically performed better than the yearly strategy (Fig 5 and S5 Fig). In years without a large outbreak, the adaptive threshold strategy performed worst, even worse than not spraying at all, because herd immunity was reduced from previous years of spraying, while no spraying happened in that year because the threshold was not met (See, for example, the 2003–2004 and 2006–2007 seasons, S5 Fig).

When we initiated spraying with a fixed threshold (i.e., one that does not depend on the mean and standard deviation from recent years), then the number of infections had a nonlinear relationship with the magnitude of the threshold (monthly threshold is shown in Fig 6, left column; weekly threshold is shown in S6 Fig and comparison of monthly and weekly in S7 Fig). Counterintuitively, we observed higher numbers of infections at lower thresholds for spraying than at higher thresholds, even though for low thresholds we sprayed more (Fig 6, top and bottom rows). We also see a linear increase in mosquito abundance as we increase the threshold and reduce the amount of spraying, although the effect on average mosquito abundance is weak due to the short-lived effect of ULV. Considering the relationship between total infections and threshold, we saw what appeared to be two regimes: declining numbers of infections as we increased the threshold for spraying until about 400 cases per month (or 130 per week), and a more modest increase in the number of infections as we increased the threshold above that number. To explore this pattern, we stratified the simulations into those for which the threshold was below 400 cases per month (or 130 cases per week), and those where the threshold was above this (Fig 7). When the threshold was low, spraying often occurred too soon, before outbreaks began in earnest, and, because we were limited to two spray campaigns per year, we effectively used up our quota by the time the outbreak occurred (Fig 7; top row). At higher thresholds, spray campaigns more closely corresponded to times when transmission was ongoing and, consequently, the subsequent incidence was lower even though fewer days spent spraying were required (Fig 6; bottom row). Essentially, as the effect of ULV is short-lived, the timing of spraying is more important than the total amount of spraying over the 11-year period, meaning a lower threshold is not better if it means spraying occurs too early. At low thresholds, there was bistability in the total number of infections. We explored this by looking separately at those simulations for which the total number of infections was above 210,000, and those for which it was below this value (S8 Fig). In both cases, we limited our analyses to simulations for which the threshold was between 125 and 400, the region in which the bistability occurs. This additional analysis indicated that the bistability was due to the stochastic nature of the model, and in particular, the extent to which the large epidemic in 2010 is mitigated.

## Targeted indoor residual spraying

The strategy for beginning city-wide TIRS that led to fewest infections was to begin when the weekly incidence was one standard deviation above the mean, resulting in 9,900 infections (IQR: 8,720–11,300). This strategy reacted more quickly than the best adaptive threshold ULV strategy (i.e., monitoring incidence monthly). The result was many fewer infections (17-fold) than the best ULV strategy, which led to a median of 172,000 infections. Because spraying the whole city once with TIRS took longer than spraying the whole city three times with ULV (35

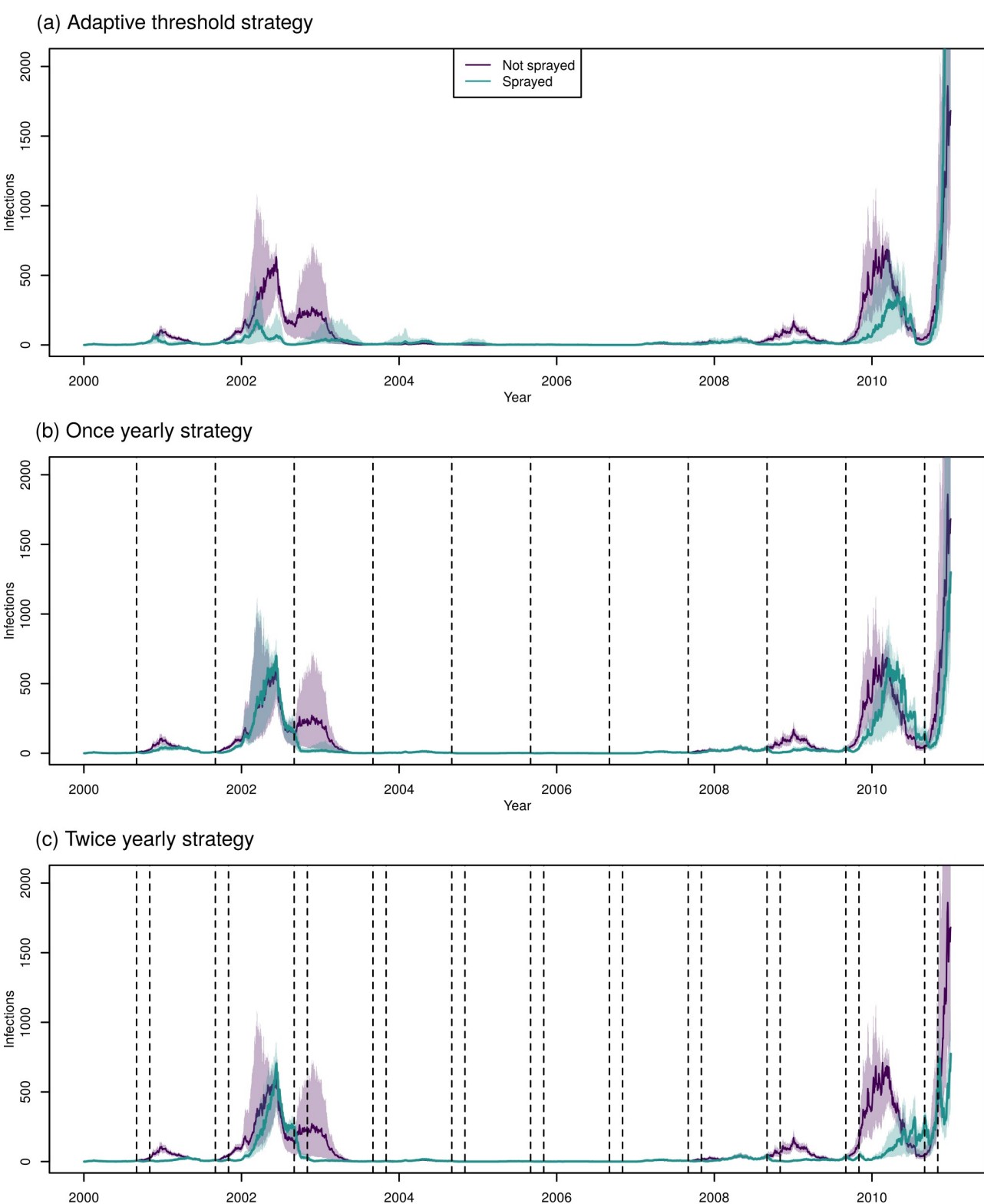

**Fig 5. Time-series of incidence of human infections for the best ULV strategy in each category.** In each plot, the green lines represent predictions without spraying, and purple represents the given strategy. The line represents the median of all 400 simulations and shading represents the inter-quartile range. The vertical dotted lines represent the start of city-wide ULV campaigns (not displayed for the threshold strategy, as in this case campaigns start at differing times depending on incidence).

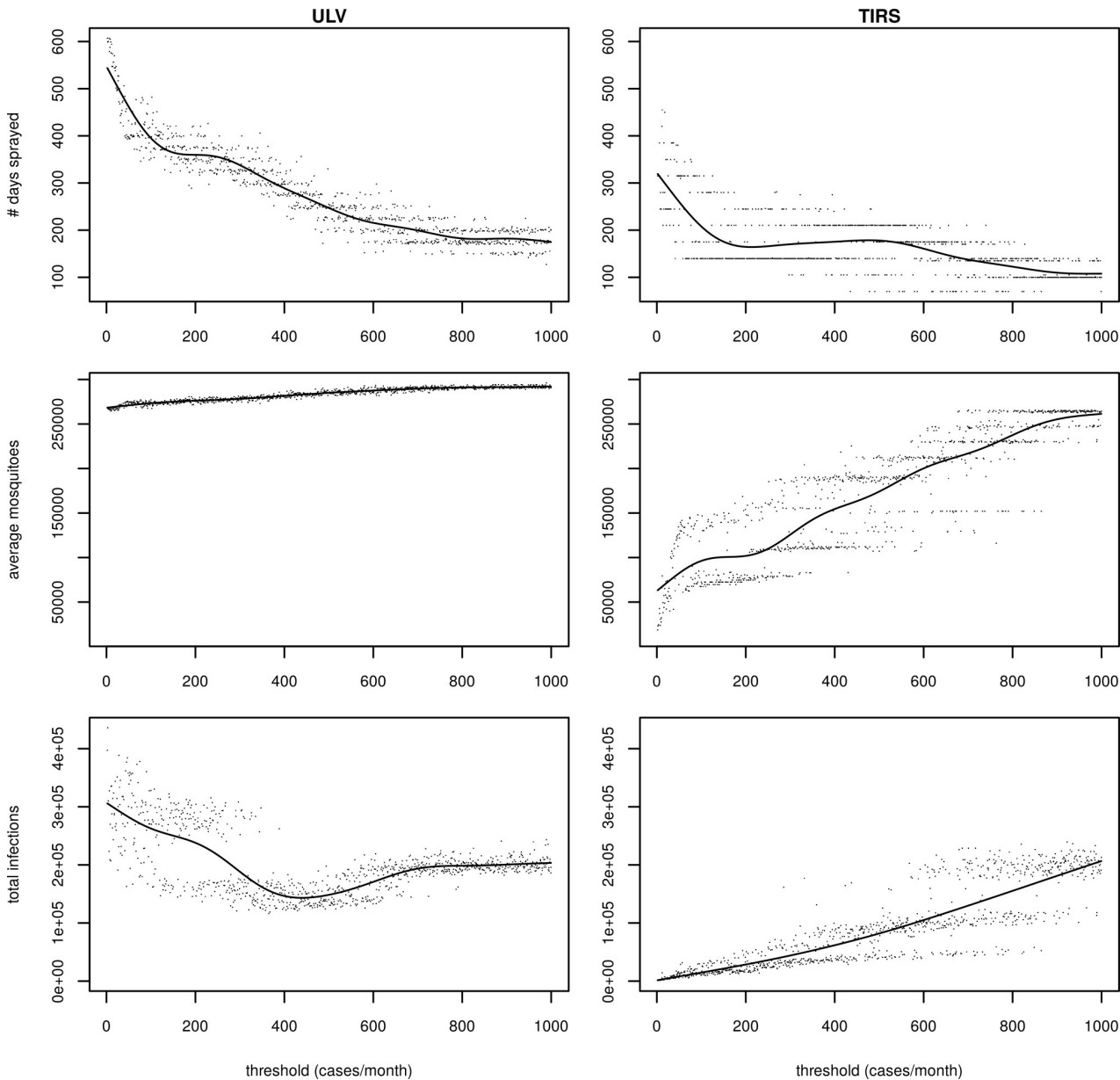

**Fig 6. Results when initiating ULV spraying according to a fixed threshold that was monitored on a monthly basis.** The top row shows the number of days spent spraying over the 11 year period, the middle row the mean mosquito abundance, and the bottom row the total number of dengue infections. The left column is ULV spraying and the right column TIRS. Each point represents one model simulation, and the line represents predictions by a fitted generalized additive model.

days vs 25 days), the yearly strategy required spraying for more days for TIRS than for ULV. When using an adaptive threshold strategy though, fewer days were spent spraying than any other strategy tested (median of 280 days), despite it also having the largest impact on number of infections (Table 2). The best yearly strategy was to begin spraying each September, which resulted in 11,800 infections (IQR: 9,750–14,100). There were smaller differences between the yearly TIRS strategies than for ULV spraying, particularly for the yearly strategies (Fig 8C). Overall, all TIRS strategies had a much larger impact on the number of infections than did ULV strategies, and were able to almost completely avert some outbreaks in later years (Fig 9).

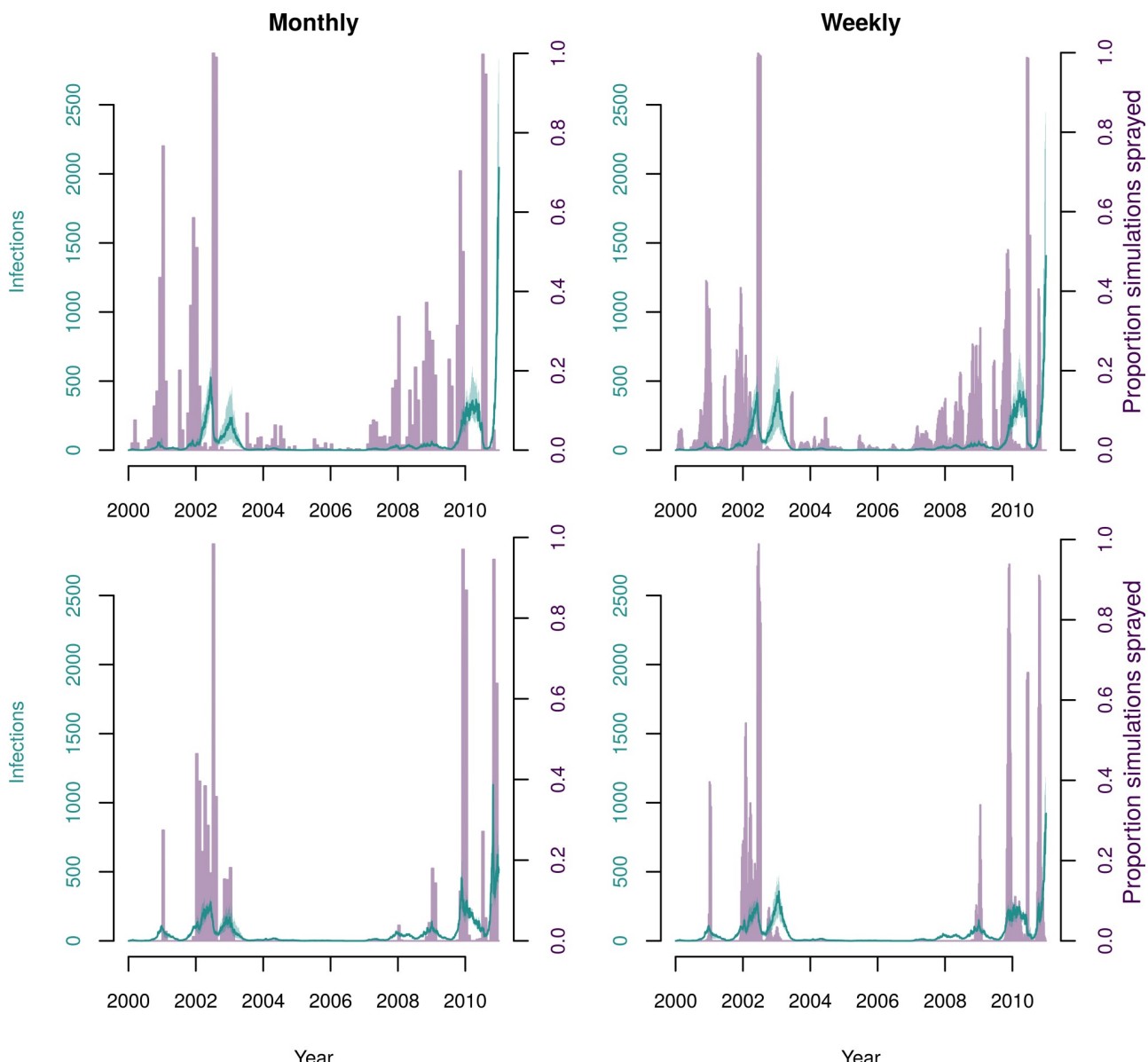

**Fig 7. Comparison of frequency of ULV spraying and human incidence with different fixed thresholds to initiate spraying.** In all panels, lilac bars represent the proportion of simulations which were undertaking spraying on the given day, the green line represents median incident infections, and the green shading represents the interquartile range. The left column shows when thresholds were monitored monthly, and the right column when they were monitored weekly. In the top row, the threshold was low (400 and below for monthly, 130 and below for weekly), and in the bottom row, the threshold was high (above 400 for monthly, and above 130 for weekly).

This is because repeated applications of IRS almost eliminate *Ae. aegypti* from Iquitos (S9 Fig). When initiating TIRS after incidence exceeded a fixed threshold, the number of predicted infections and the average mosquito abundance increased approximately linearly with the threshold (Fig 6, right column). Unlike for ULV, increasing the threshold for TIRS initiation tends to increase the number of cases. This is because timing is less important for TIRS, so spraying earlier in the season is not as big of a disadvantage as for ULV. For thresholds above 500 cases per month, there was a tristability in the total number of infections. To explore this, we stratified the simulations that had a threshold above 500 into three groups based on the

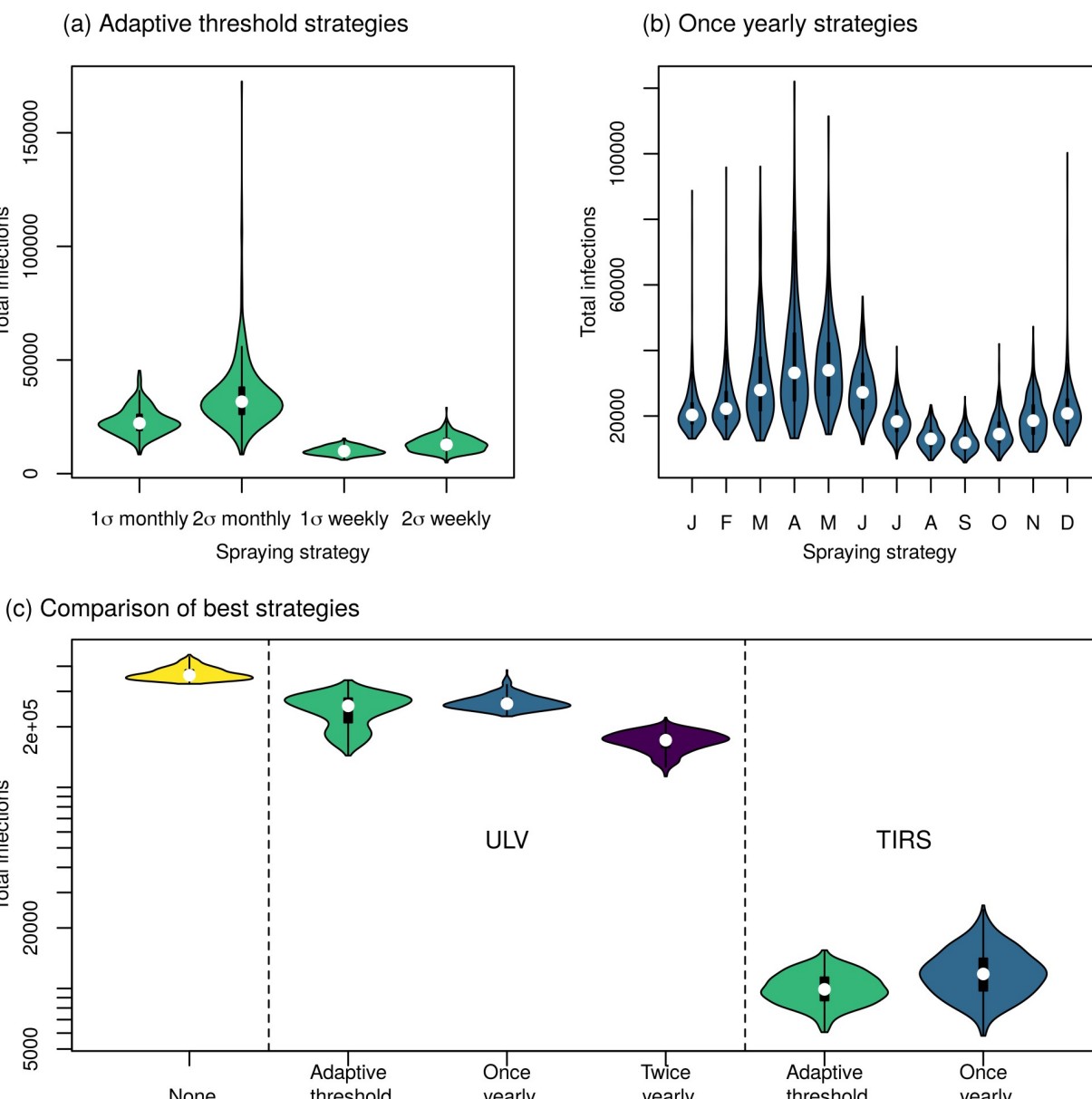

**Fig 8. Predicted human infections following city-wide TIRS spraying.** (a) Comparison of adaptive threshold strategies for initiating spraying–spraying began when the monthly or weekly incidence was one or two standard deviations above the mean for that period from the last five years, as shown on the x-axis. (b) Comparison of yearly city-wide spraying strategies, began on the first day of the indicated month. (c) Comparison of the best strategies in each category of both ULV and TIRS: for ULV, adaptive threshold corresponds to starting when monthly incidence was more than one standard deviation above the mean, once yearly corresponds to spraying in September, and twice yearly corresponds to spraying in September and November; for TIRS, adaptive threshold corresponds to starting when weekly incidence is more than one standard deviation above the mean, and once yearly corresponds to spraying in September.

number of infections (S10 Fig). Similar to the bistability observed for ULV spraying, the difference between the simulations with intermediate number of infections and those with high number of infections was due to whether the large epidemic in 2010 was averted or not. Simulations with low total number of infections were due to total elimination of *Ae. aegypti* from

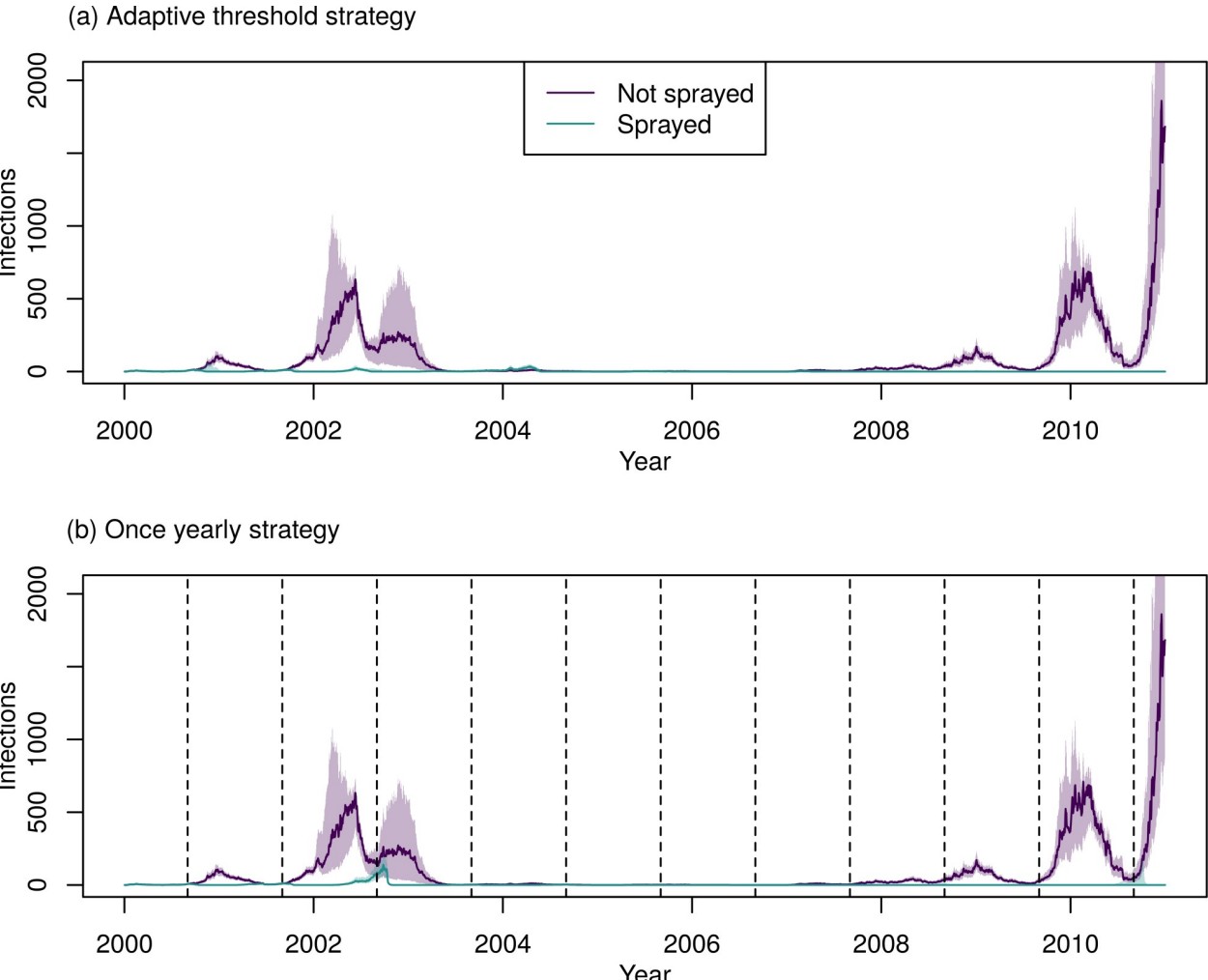

**Fig 9. Time-series of incidence of human infections for the best TIRS strategy in each category, for (a) an adaptive strategy and (b) a yearly strategy.** In each plot the purple lines represent the predictions without spraying, and the green represents the given strategy. The line represents the median of all 400 simulations and the shading represents the inter-quartile range. The vertical dotted lines represent the start of city-wide TIRS campaigns (not displayed for the threshold strategy as in this case campaigns start at differing times depending on incidence).

Iquitos in 2005 (S10 Fig). Hence, in those simulations, we observed no further dengue virus transmission following the elimination.

## Sensitivity analysis

Keeping all other parameters the same, we jointly varied the thoroughness of spraying (i.e., the increase in the daily mortality rate), the compliance of houses, and, in the case of ULV strategies, the delay between spray cycles. For the best adaptive threshold strategy, surveillance effort did not have a large effect on the predicted number of infections. This is because our threshold is based on incidence from recent years, so if only a proportion of cases are notified, then the threshold will be that proportion of its value if all cases were notified, and the time at which spraying starts will be similar. Increasing the thoroughness of the spraying (or, equivalently, the efficacy of the treatment) leads to fewer infections averted for small values of thoroughness (Fig 10, left column). However, for values of thoroughness above 3 (a daily mortality risk of

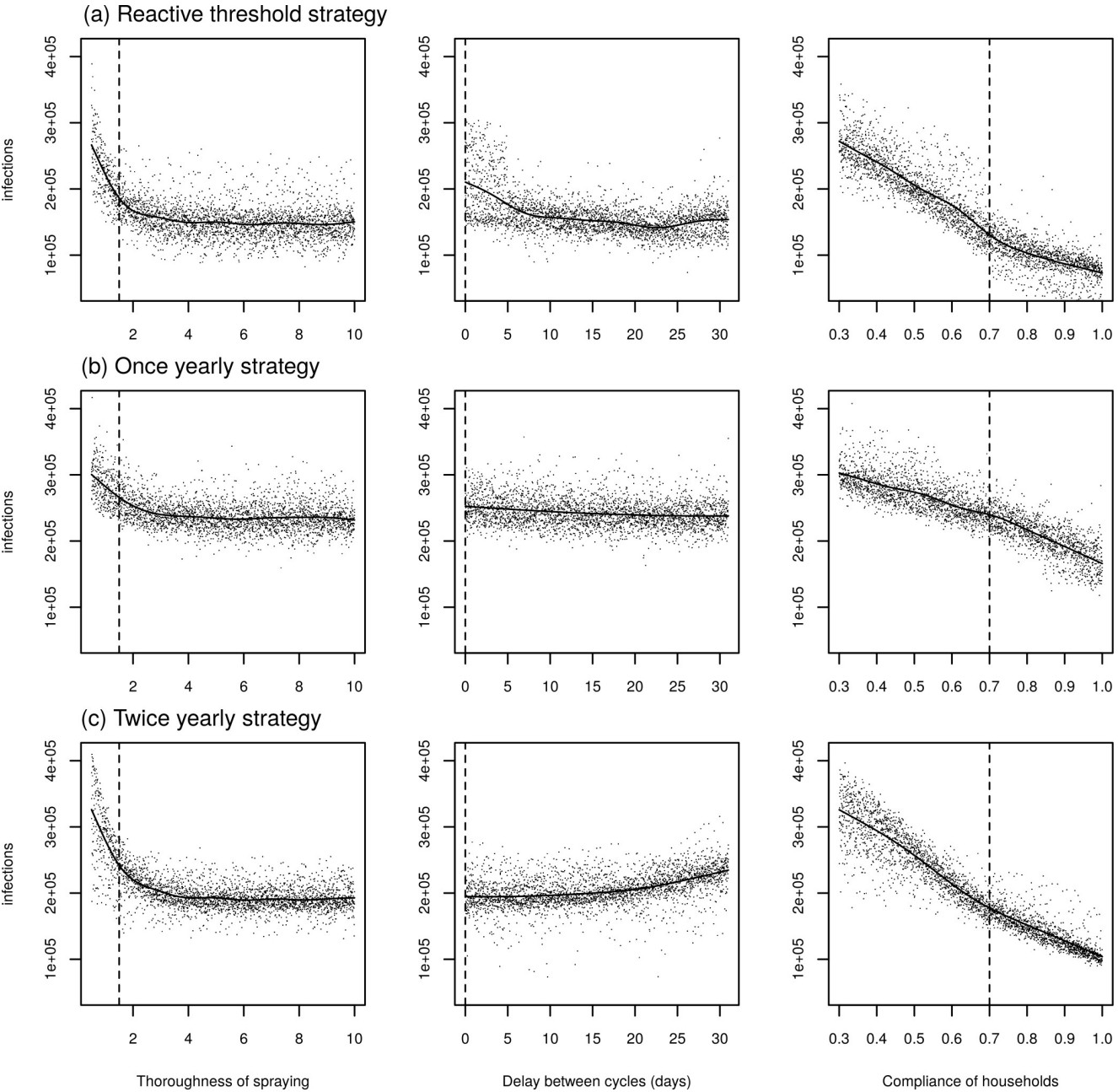

**Fig 10.** Sensitivity analysis jointly varying the thoroughness of ULV spraying (i.e., the increase in mosquito mortality rate on the day of spraying) (left column), the delay, in days, between cycles (middle column), and the compliance of households. The first row shows the results using the best adaptive threshold strategy (starting spraying when the monthly incidence is more than one standard deviation above the mean for that month for the last five years), the second row using the best yearly strategy (spraying in September) and the bottom row the best twice yearly strategy (spraying in September and November). Each point represents a model simulation, and the line represents a fitted generalized additive model. Vertical dashed lines show the value used in the baseline simulation.

about 95%), increasing the thoroughness further does not lead to further gains. The probability that a household complies with spraying has a strong negative relationship with the number of infections predicted for all strategies (Fig 10, right column).

In the case of TIRS, reducing the thoroughness of spraying did not have a strong effect on the number of infections (S11 Fig). Even for low thoroughness of 0.5, all TIRS implementations lead to fewer than 80,000 infections, a more than 4.5-fold reduction from the baseline number of infections (361,000). For the adaptive threshold strategy, reduced household compliance did not have a large effect on the number of infections above a compliance of 0.4 (much below the observed compliance of 0.7). Compliance had a stronger impact on the number of infections for the yearly strategy, in which improved compliance generally led to fewer infections. For both TIRS and ULV spraying campaigns, the variance in compliance determined more than half of the overall variance in the number of infections in all cases except for the adaptive threshold TIRS strategy (S12 Fig). In the case of adaptive threshold TIRS, there was likely a substantial interaction between compliance and the thoroughness of the spraying, with the interaction between these terms determining up to 39% of the variance in the output, although it should be noted that only the first-order effects are accurate for stochastic models [65].

When we reduced thoroughness of TIRS to 1.5 (i.e., the same as that used in ULV spraying), then the best reactive threshold strategy was the same as when the thoroughness was 9: spraying when weekly incidence is more than one standard deviation above the mean. Moreover, it had only a small impact on the number of infections we predicted: 12,800 infections (IQR: 12,000–14,100) compared to 9,900 in the baseline case. When we reduced thoroughness further (to 0.75), the best threshold strategy remained the same, and the impact of the number of infections was still large: we predicted 14,300 infections (IQR: 13,500–15,700). In the case of the once yearly strategy, reducing the thoroughness to 1.5 or 0.75 led to predictions of 35,800 (IQR: 32,500–38,500) and 45,300 (IQR: 41,900–49,700), respectively, and changed the best month to spray to August (compared to September in the baseline case).

In the baseline case, we assumed that 100% of symptomatic cases were notified when calculating whether the threshold for response was reached. However, as the adaptive threshold was itself based on past incidence of notified cases, it indirectly included reporting rate, and so a fixed level under-reporting should not greatly affect response timing, beyond increasing stochasticity. We tested this logic by decreasing the reporting rate to 10%. In this case we observed 248,000 (IQR: 197,000–275,000) and 6,370 (IQR: 5,030–7,620) infections for city-wide ULV and IRS campaigns, respectively (recall this is the total number of infections and so is unaffected by reporting rate). These values were both below the baseline median prediction for full reporting. The best strategies in both instances remained the same as in the baseline, full-reporting case. If surveillance of cases lagged by two weeks, the best adaptive strategies remained the same for both IRS and ULV. For TIRS we observed slightly more infections than when there was no lag (12,100, IQR: 10,300–14,500), whereas for ULV we actually observed many fewer infections compared to when there was no lag (164,000, IQR: 156,000–172,000).

## Discussion

### Summary of main findings

We used an agent-based model of DENV transmission in Iquitos, Peru, to compare strategies for initiating city-wide spraying with either ULV or TIRS. None of the city-wide ULV spraying strategies were able to prevent outbreaks. The best strategies reduced the total number of infections over an eleven-year period by around a half. Strategies that used TIRS were able to almost completely eliminate *Ae. aegypti* from Iquitos, and so prevent an order of magnitude more infections than ULV. The best strategy for ULV spraying was to spray twice per year, in September and November. Spraying yearly in September prevented slightly fewer infections, but required spraying slightly less, than the best adaptive threshold strategy. The yearly and twice

yearly strategies also tended to lead to fewer infections than the adaptive threshold strategies in those years when there was not a large outbreak. The best strategy tested for TIRS was an adaptive threshold, which had the biggest impact on the numbers of infections of all strategies tested. Moreover, it also required the fewest days spent spraying compared to all other strategies (280 days over 11 years).

## Explanation and interpretation of results

When considering initiation of a city-wide ULV campaign, two factors stood out as optimizing the impact of outbreak response: (1) begin spraying when the monthly incidence is one standard deviation above the mean, and (2) use a relatively high fixed threshold (for Iquitos: ~400 cases/month) for initiating outbreak response. Taken together, these observations indicate that after an initial increase in incidence, not reacting too quickly can result in a more effective city-wide ULV response. This makes sense, due to the short-term effect of ULV spraying and the capacity of *Ae. aegypti* populations to rebound rapidly. If we instead consider TIRS, our results indicate that timing is less important, due to the residual effect of the insecticide. Moreover, reductions in vector abundances and numbers of human infections would be even further reduced with longer lasting insecticides, such as those with 150-day effects that are now becoming available [45].

Our sensitivity analysis indicated that the level of surveillance effort did not have a strong effect on the predicted number of infections, due to the fact that the adaptive threshold calculation inherently captures this under-reporting, if it occurs at a constant rate. This would not be the case, however, if the rate of under-reporting changed over time. For instance, if reporting rate increased through time, then thresholds would be based on a smaller proportion of cases than the current year's incidence, which would lead to us spraying too soon due to an artificially small threshold. This could happen if, for example, reporting of DENV cases became more frequent as awareness of symptoms grew among the public and/or clinicians [66]. In addition, a lag in reporting did not have a big impact on our model's results about the number of infections prevented by a city-wide TIRS campaign. This was not the case for ULV, though. In that case, a lag of two weeks actually led to fewer infections, implying that, if there is not an inherent lag in reporting, it may be worthwhile to wait once the threshold has passed before beginning a city-wide ULV campaign.

Reassuringly, the impact of a city-wide TIRS campaign was robust to more pessimistic assumptions about the thoroughness with which the insecticide is sprayed (or its efficacy) and the compliance of households, in addition to under-reporting and lags in reporting. This means that, even when the increase in mosquito mortality caused by TIRS at baseline was an order of magnitude below that observed by Dunbar et al. [25], or when half as many houses were treated as observed in city-wide ULV campaigns, then the effect of a city-wide IRS campaign was not greatly impacted.

Because the seasonality of DENV transmission is highly irregular in Iquitos [22], a characteristic of DENV transmission in general, yearly strategies can be expected to perform very well in some years (e.g., 2002–03) but poorly in others (e.g., 2001–02), especially if spraying occurs too soon. This implies that caution should be taken to not overinterpret our result that September seemed to be the best month to initiate ULV spraying. September was largely best because application at that time strongly mitigated the 2002–2003 season, which was specific to the particular importation patterns that sparked local transmission during the 2000–2010 period of our analysis. Generalizing across ULV strategies though, it seems that strategies that initiate just before or early in the season perform best.

## Comparison to the other studies

Our prediction that the optimal indoor ULV strategy could lead to a reduction of infections by up to around 50% exceeds an estimate from a recent modeling study from Porto Alegre, Brazil, which reported that outdoor truck-mounted ULV spraying reduced the number of secondary infections by around 25% [10]. At the same time, our estimate is lower than the 85% reduction in infections due to outdoor ULV spraying reported by Wahid et al. [67] in Malakar, Indonesia, although ULV was applied there in conjunction with other interventions (reactive ULV, larviciding, and larval source reduction) [67]. A previous modeling study of IRS spraying in Merida, Mexico found that proactive strategies (i.e., before the season) outperformed reactive strategies [45]. That is commensurate with our results, as each of our threshold strategies and the best performing yearly strategies all involve spraying before the high transmission season. The observation of Hladish et al. [45] that campaigns that start after the peak in incidence can still have a large effect is consistent with our result that the month when we started TIRS was less important than for ULV. Our prediction of a 97% reduction in the number of infections for repeated TIRS campaigns with about 70% coverage each year is greater than that found by Haldish et al. [45] (79% reduction in infections over 5 years with 75% coverage in Merida, Mexico) and Vazquez-Prokopec et al. [24] (86% reduction in transmission in treated houses in Cairns, Australia). Comparing to Hladish et al. [45], our higher predicted impact may be because our model incorporates a more detailed entomological component with immature stages and spatial heterogeneity. This means that feedbacks caused by fewer mosquitoes laying fewer eggs, and stochastic local fade-out of adult mosquitoes, allow the *Ae. aegypti* population to be reduced to very low numbers after several years of TIRS application.

For all ULV strategies, the approach that led to the fewest infections was not necessarily the same as the strategy that reduced mosquito abundance the most, reemphasizing the point made by previous studies of the importance of measuring epidemiological endpoints when assessing vector control [68,69]. This difference is likely due to DENV transmission being the result of a complex interplay of factors, not simply a direct, positive relationship with *Ae. aegypti* abundance. The best strategy also differed by year. In years with low incidence, the adaptive threshold strategies performed poorly, because a response was not triggered.

## Limitations and strengths

Although we have compared results of city-wide ULV spraying with results of city-wide TIRS, there are some caveats to this comparison. First, we parameterized ULV spraying using a study of actual city-wide spraying campaigns in Iquitos, while we parameterized TIRS using a controlled study from a different country. It is possible that in reality TIRS may have lower effectiveness, although we saw in our sensitivity analysis that if we reduced TIRS to the level of thoroughness used for ULV, the impact on the number of infections was still much greater for TIRS. Secondly, it may not be feasible to undertake city-wide TIRS campaigns, due to the greater cost and time commitment associated with city-wide spray campaigns. On the other hand, our results show that, in the long-term, we would actually need to spray less using TIRS due to the reduced number of infections and large reduction in mosquito abundance.

A limitation of our study is that there are few published results on the impact of ULV or TIRS on dengue incidence with which to validate our model [68,69]. This is mitigated somewhat by our model's incorporation of mosquito population dynamics that match observed patterns from Iquitos [52], as well as matching ULV mortality effects to a study carried out in Iquitos and IRS effects to a controlled study in Mexico [11,12]. A second limitation is that, because our model reproduces the seasonal patterns observed in the period 2000–2010, our results may be somewhat specific to DENV transmission and mosquito population patterns at

that particular place and time. While our results regarding threshold spraying strategies are likely to be robust to this this concern, our predictions for when regular spraying should begin may be less robust, and could, for example, differ if importation rates peak at different points in the transmission season. A third limitation is that the force of importation and magnitude of mosquito abundance is calibrated to an estimate of the force of infection, which includes spraying events that took place in 2000–2010. This may mean that we overestimated the effect of spraying, as our baseline in effect already includes spraying. On the other hand, grounding of our model in data from Iquitos, which is an extremely well-characterized site for DENV transmission and *Ae*. a*egypti* population dynamics, is a notable strength. Another strength is our model's level of detail, which is something that enables us to capture the interplay of two important feedbacks in mosquito population dynamics following spraying: (1) density-dependent mortality in the larval stage causing the population to rebound and (2) reduced egg-laying by adults. We are also able to capture local mosquito population depletion to zero due to demographic stochasticity and subsequent population rebounding due to mosquito movement.

## Implications and next steps

Our results indicate that the city-wide ULV and TIRS campaigns would have reduced the number of DENV infections in Iquitos by up to half relative to the baseline scenario that we modeled. Although a well-timed campaign could be expected to mitigate transmission in a particular season, it would be difficult to prevent an outbreak altogether using ULV or TIRS. Similarly, selecting a single strategy that consistently mitigated outbreaks across multiple years proved to be difficult. For example, our adaptive threshold strategies performed well during the 2001–2002 transmission season, but poorly in 2002–2003. The opposite was true of the yearly strategy. With indoor ULV spraying, the best strategy was with a fixed threshold of around 400 cases per month. A downside of this approach is that it requires accurate, timely, and potentially expensive surveillance. More field work is needed to better understand the feasibility and effectiveness of city-wide TIRS, including its spatially targeted application in combination with ULV. We predict, however, that city-wide TIRS, if feasible, will have a greater impact than ULV without asking significantly more from surveillance.

## Supporting information

**S1 Text. Further details of the agent-based model of dengue virus transmission.**
(ODT)

**S1 Fig.** Monthly, serotype-specific incidence of infection per capita, as estimated by Reiner et al. [51](gray bands) and as reproduced by our calibrated model (colored bands).
(TIF)

**S2 Fig.** Median numbers of infections on a monthly basis for each serotype, stratified by whether the infection was acquired through biting by an infectious mosquito (colored) or by exogenously driven infections (gray) that were used to seed transmission in the model.
(TIF)

**S3 Fig. Median proportion of the population immune to each dengue serotype.** Dark gray indicates permanent homologous immunity, light gray indicates temporary heterologous immunity, and colored regions indicate susceptibility.
(TIF)

**S4 Fig. Predicted average mosquito abundance following city-wide ULV spraying.** (a) Comparison of reactive strategies for initiating spraying; spraying began when the monthly or weekly incidence was one or two standard deviations above the mean for that period from the last five years, as shown on the x-axis. (b) Comparison of yearly city-wide spraying strategies, beginning on the first day of the shown month. (c) Comparison of the median predicted cases for twice yearly spraying strategies, beginning on the first days of the shown month. Darker colors correspond to fewer cases, and the diagonal shows yearly spraying strategies. (d) Comparison of the best strategies in each category: adaptive threshold corresponds to starting when monthly incidence was more than one standard deviation above the mean, once yearly corresponds to spraying in September, twice yearly corresponds to spraying in September and November.
(TIF)

**S5 Fig. Predicted number of infections following city-wide ULV spraying, by season.** Each figure compares the best strategies in each category, for that season.
(TIF)

**S6 Fig. Results when initiating ULV spraying according to a fixed threshold that is monitored on a weekly basis.** The top row shows the number of days spent spraying over the 11 year period, the middle row the mean mosquito abundance, and the bottom row the total number of dengue infections. The left column is ULV spraying and the right column TIRS. Each point represents one model simulation, and the line represents predictions by a fitted generalized additive model.
(TIF)

**S7 Fig.** Results when initiating ULV spraying according to a fixed threshold that is monitored on a monthly (purple) or weekly (teal) basis. The top row shows the number of days spent spraying over the 11 year period, the middle row the mean mosquito abundance, and the bottom row the total number of dengue infections. The left column is ULV spraying and the right column TIRS. Each point represents one model simulation, and the line represents predictions by a fitted generalized additive model.
(TIF)

**S8 Fig.** Timeseries showing median and 90% confidence intervals of simulations in different regions of the bistability observed in Fig 6 (left column). All simulations include ULV spraying with a fixed threshold between 125 and 400 cases/month. Simulations marked "high" (purple) are those for which the total number of infections was above 210,000, and "low" are those below this value. Panels (b) and (c) show detail of panel (a).
(TIF)

**S9 Fig.** Time-series of mosquito abundance for the best TIRS strategy for (a) the best adaptive threshold strategy (starting when weekly incidence exceeds the mean by $1\sigma$) and (b) the best yearly strategy (starting in September). In each plot the purple lines represent the predictions without spraying, and the green represents the given strategy. The line represents the median of all 400 simulations and the shading represents the inter-quartile range.
(TIF)

**S10 Fig. Timeseries showing median and 90% confidence intervals of simulations in different regions of the tristability observed in Fig 6 (right column).** All simulations include TIRS spraying with a fixed threshold above 500 cases/month. Simulations marked "high" (purple) are those for which the total number of infections was above 150,000, "medium" (teal) are those between 75,000 and 150,000 and "low" (yellow) are those below 75,000. The top panel

shows number of infections and the bottom panel mosquito abundance.
(TIF)

**S11 Fig. The effect of varying household compliance and thoroughness of spraying on the median number of infections for the best adaptive threshold strategy with TIRS (spraying when monthly incidence exceeds one standard deviation above the mean).** The vertical line shows the value used in the baseline simulations. The solid line represents a fitted multivariable generalized additive model.
(TIF)

**S12 Fig.** Pie charts showing the proportion of variance in the output that is explained by variance in the sampled input parameters for (a) adaptive threshold ULV strategies, (b) yearly ULV strategies, (c) twice yearly ULV strategies, (d) adaptive threshold TIRS strategies, and (e) yearly TIRS strategies. Higher-order terms include interactions between parameters as well as aleatory uncertainty; in the case of TIRS the only interaction is the interaction between compliance and thoroughness as only two parameters were varied.
(TIF)

## Acknowledgments

We thank the residents of Iquitos for their participation in this study. We greatly appreciate the support of the Loreto Regional Health Department, including Drs. Hugo Rodriguez-Ferruci, Christian Carey, Carlos Alvarez, Hernan Silva, and Lic. Wilma Casanova Rojas who all facilitated our work in Iquitos. We thank the NAMRU-6 Virology and Emerging Infections Department (VEID) and Entomology Department leadership who provided institutional support, IRB guidance and support supervising field staff during the years 2000–2010 when the data used in these models was collected. We also appreciate the careful commentary and advice provided by the NAMRU-6 IRB and Research Administration Program for the duration of this study. We thank the NAMRU-6 VEID field teams provided who daily support through duration of the project and without whom the capture of acute dengue cases would not have been possible. In particular we thank Gabriela Vasquez de la Torre for her administrative support for the project.

The views expressed in this article are those of the authors and do not necessarily reflect the official policies or positions of the Department of the Navy, Department of Defense, nor the U. S. Government.

## Copyright statement

ACM and HA were employees of the United States government. This work was prepared as part of their official duties. Title 17 U.S.C. 105 provides that 'copyright protection under this title is not available for any work of the United States Government.' Title 17 U.S.C. 101 defines a U.S. Government work as work prepared by a military service member or employee of the U. S. Government as part of that person's official duties.

## Author Contributions

**Conceptualization:** Gonzalo M. Vazquez-Prokopec, Thomas W. Scott, Robert C. Reiner Jr., T. Alex Perkins.

**Data curation:** Helvio Astete, Amy C. Morrison.

**Formal analysis:** Sean M. Cavany, Guido España, Robert C. Reiner Jr., T. Alex Perkins.

**Funding acquisition:** Thomas W. Scott, T. Alex Perkins.

**Investigation:** Sean M. Cavany, Guido España, T. Alex Perkins.

**Methodology:** Sean M. Cavany, Guido España, T. Alex Perkins.

**Project administration:** Gonzalo M. Vazquez-Prokopec, Thomas W. Scott, T. Alex Perkins.

**Resources:** T. Alex Perkins.

**Software:** Sean M. Cavany, Guido España, Robert C. Reiner Jr., T. Alex Perkins.

**Supervision:** T. Alex Perkins.

**Validation:** Sean M. Cavany.

**Visualization:** Sean M. Cavany, T. Alex Perkins.

**Writing – original draft:** Sean M. Cavany.

**Writing – review & editing:** Sean M. Cavany, Guido España, Alun L. Lloyd, Lance A. Waller, Uriel Kitron, Helvio Astete, William H. Elson, Gonzalo M. Vazquez-Prokopec, Thomas W. Scott, Amy C. Morrison, Robert C. Reiner Jr., T. Alex Perkins.

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
