## [Decision Letter · Decision Letter 0]

3 Dec 2019

Dear Dr Cavany,

Thank you very much for submitting your manuscript, 'Optimizing the deployment of ultra-low volume and indoor residual spraying for dengue outbreak response', to PLOS Computational Biology. As with all papers submitted to the journal, yours was fully evaluated by the PLOS Computational Biology editorial team, and in this case, by two independent peer reviewers.

Both reviewers appreciated the contribution of this work; they differed however on their comments regarding clarity and completeness of the presentation. They identified some aspects of the manuscript that should be improved, in particular referee 2. 

We would therefore like to ask you to modify the manuscript according to the review recommendations before we can consider your manuscript for acceptance. Your revisions should address the specific points made by each reviewer and we encourage you to respond to particular issues Please note while forming your response, if your article is accepted, you may have the opportunity to make the peer review history publicly available. The record will include editor decision letters (with reviews) and your responses to reviewer comments. If eligible, we will contact you to opt in or out.raised.

- Supporting Information uploaded as separate files, titled 'Dataset', 'Figure', 'Table', 'Text', 'Protocol', 'Audio', or 'Video'.

We hope to receive your revised manuscript within the next 30 days. If you anticipate any delay in its return, we ask that you let us know the expected resubmission date by email at ploscompbiol@plos.org.

Sincerely,

Mercedes Pascual

Associate Editor

PLOS Computational Biology

Virginia Pitzer

Deputy Editor

PLOS Computational Biology

[LINK]

Reviewer's Responses to Questions

**Comments to the Authors:**

Reviewer #1: Summary

Application of ultra-low volume spraying remains the primary intervention strategy against dengue. The authors demonstrate that targeted indoor residual spraying can more effectively control dengue using a mathematical model calibrated to data from Iquitos, Peru.

I thoroughly enjoyed reading this paper and would recommend it for publication given the following (very) minor revisions aimed at clarifying figures and explaining model dynamics.

Minor points:

1. The first citation in the paper on line 64 is inappropriate for the statement. The cited paper only presents several outbreaks in one Brazilian. Please can the authors consider a different reference giving an idea of rising global dengue incidence.

2. Although most axis labels in Figures 3, 4 and 7 are the same, please add axis labels.

3. For Figure 3c, please can the authors consider adding a colourbar legend so the reader can more easily compare total predicted human infections with other subfigures.

4. The bi-stable behavior of the total number of infections at low thresholds (around 200 cases per month) with ULV spraying (Figure 5, bottom panels) is interesting. Is this due to the stochasticity of the model? I.e. in some simulations is the quota used up before a large outbreak, but in other simulations is there just enough sprays remaining to avert these infections?

5. Similarly, can the authors explain the tri-stable dynamics of the total number of infections at large thresholds (600 cases per month and above) with TIRS.

Reviewer #2: the review is uploaded as an attachment

**Have all data underlying the figures and results presented in the manuscript been provided?**

Reviewer #1: Yes

Reviewer #2: Yes

PLOS authors have the option to publish the peer review history of their article (what does this mean?). If published, this will include your full peer review and any attached files.

Reviewer #1: No

Reviewer #2: No

---

## [Editor Report · Decision Letter 1]

24 Feb 2020

Dear Mr. Cavany,

We are pleased to inform you that your manuscript 'Optimizing the deployment of ultra-low volume and indoor residual spraying for dengue outbreak response' has been provisionally accepted for publication in PLOS Computational Biology.

Before your manuscript can be formally accepted you will need to complete some formatting changes, which you will receive in a follow up email. A member of our team will be in touch within two working days with a set of requests.

Best regards,

Mercedes Pascual

Associate Editor

PLOS Computational Biology

Virginia Pitzer

Deputy Editor

PLOS Computational Biology

---

## [Editor Report · Acceptance letter]

15 Apr 2020

PCOMPBIOL-D-19-01721R1 

Optimizing the deployment of ultra-low volume and targeted indoor residual spraying for dengue outbreak response

Dear Dr Cavany,

I am pleased to inform you that your manuscript has been formally accepted for publication in PLOS Computational Biology. Your manuscript is now with our production department and you will be notified of the publication date in due course.

With kind regards,

Sarah Hammond
